# Tool for Observing Play Outdoors (TOPO): A New Typology for Capturing Children’s Play Behaviors in Outdoor Environments

**DOI:** 10.3390/ijerph17155611

**Published:** 2020-08-04

**Authors:** Janet Loebach, Adina Cox

**Affiliations:** 1Design and Environmental Analysis, Cornell University, Ithaca, NY 14850, USA; 2Landscape Architecture, University of Kentucky, Lexington, KY 40506, USA; Adina.Cox@uky.edu

**Keywords:** outdoor play, play types, typology, play environments, children’s behavior, play spaces, children’s development, observation tool, nature play

## Abstract

Engagement in play has been definitively linked to the healthy development of children across physical, social, cognitive, and emotional domains. The enriched nature of high-quality outdoor play environments can afford a greater diversity of opportunities for play than indoor settings. To more effectively design outdoor play settings, we must better understand how the physical environment supports, or hinders, the different types of play which suit children’s needs and interests. However, play typologies or observation tools available to date do not adequately capture the unique characteristics of outdoor play. This paper outlines the development and testing of the Tool for Observing Play Outdoors (TOPO), a new typology of outdoor play, as well as a systematic field observational protocol which can be used to effectively depict children’s behaviors in outdoor spaces, as well as evaluate the play environment itself. The tool can be deployed in either a collapsed or expanded form to serve the needs of a wide range of studies and environments. This new tool represents a significant advance in the ability to fully and effectively study and plan outdoor play environments to provide more diverse, high-quality play settings that will support the healthy development of children across the spectrum.

## 1. The Developmental Drive for Play

Children’s play activities, so long considered to be largely superfluous and aimless [1], or as a way for children to let off steam between formal learning activities, have been definitively linked to the health and development of children [1,2]. Several reviews of the research evidence conducted by the American Academy of Pediatrics (APA) clearly demonstrate the “critical importance of play in facilitating parent engagement; promoting safe, stable and nurturing relationships; encouraging the development of numerous competencies, including executive function skills; and improving life course trajectories” [3,4,5] (p. 2). Through play, children learn how to engage successfully in their socio-cultural environments, creating and trying out experiences in safe simulations which do not threaten their physical or emotional well-being [2,3]. The impact of play is cumulative and iterative; skills and knowledge accrue during play, nurturing both short- and long-term effects on children’s development [1,2]. Brown claims that play seems to be “so important to our development and survival that the impulse to play has become a biological drive… the impulse to play is internally generated” [2] (p. 42). Children are inherently drawn to the low-risk scenarios of play in order to learn, grow, adapt, and thrive.

Neuroscience studies in recent years have also tied play in childhood to the development and proper functioning of the brain. In a recent APA review, Yogman et al. concluded from this evidence that “play is not frivolous, it is brain building” [3] (p. 5). Animal play researchers who have extensively studied the impact of play on brain development propose that during play, the brain is actually “making sense of itself through simulation and testing. Play activity is actually helping to sculpt the brain” [2] (p. 34). This helps to explain why play is most prevalent during childhood, the most significant period of brain development [2,3].

Neuroscience studies have also demonstrated that when children play, all areas of the brain “light up”, leading to adaptive and prosocial changes at each of the molecular, cellular, and behavioral levels, reinforcing that play activities can simultaneously foster development and learning across all domains—physical, social, cognitive, and emotional [2,3]. These insights suggest we should guard against an approach to play which creates artificial silos between developmental domains, associating certain types of play with a single area of development, but rather understanding play as activities which integrate development across domains. This research on the integrative impact of play on development also reminds us to acknowledge the cognitive and socioemotional aspects of playful activities; a common perception of play—particularly outdoor play—is that it is primarily physical and highly active in nature, yet quieter and more reflective activities are prevalent forms of play and important sources of learning and psychological development. One key to producing play-rich environments for children is understanding how environments can facilitate all forms of developmentally supportive activities and interactions.

## 2. Importance of Supportive Outdoor Play Environments

The forms of play in which children engage are incredibly diverse, and highly tied to this evolving developmental drive and the skills that the child has previously mastered [6,7]. The great benefit that outdoor play can offer is a greater diversity of opportunities for rich, integrated play. The more varied and less structured nature of outdoor play environments, and the fewer constraints they place on the type and range of children’s play activities, typically provide a more “enriched environment” than indoor settings, and are more likely to stimulate creativity and problem-solving, support children’s physical development, and facilitate learning [8] (p. 48). Reviewing decades of studies, Frost et al. [9] (p. 292) concluded that “children play differently outdoors than they play indoors”; play and language tends to be more complex, more dramatic play is observed, and children engage in more physically active play. Studies have demonstrated that environments which support diverse types of play will see not only a greater overall proportion of children engaged, but for longer periods of time [6]. Outdoor environments which also include substantial natural elements and materials can offer even more play opportunities; nature-based play spaces have been shown to foster more varied and complex play than traditional outdoor playgrounds [10,11,12]. A growing evidence base suggests nature-rich play spaces may also improve children’s physical, emotional, and social health by fostering motor skills development [13], reducing stress [14,15], mitigating symptoms of ADD/ADHD [16], and decreasing prevalence of allergies [17].

While child development specialists, early educators, and now neuroscientists have clearly established play as critical to human development, and largely debunked the notion that play is less important to development and learning than more formal educational activities [1,9], widespread reductions in school recess time in the U.S. and in free time for outdoor play for children in many developed countries suggest that it remains critical that we establish a clear link between play in rich, varied outdoor environments and children’s full development. Part of this effort entails generating a larger evidence base linking features and conditions of outdoor play environments to children’s healthy development.

## 3. Rationale for the Development of a New Outdoor Play Observation Tool

The study of outdoor play is important for understanding how we can more effectively design play environments that support children’s development and interests. Understanding how the physical environment supports, or hinders, different types of play can facilitate the design and programming of outdoor spaces which better suit children’s play needs and interests. However, play typologies or observation tools available to date do not adequately capture the unique characteristics of outdoor play to support effective evaluation of children’s activities or the outdoor play environment itself.

Many studies to date have utilized a play typology to help categorize children’s play behavior in relation to an outdoor environment [13,14,18,19,20]. These studies often use a previously developed play type scale (or an adaption thereof), the most prevalent of which are Frost’s *Play Observation Form* [21], Hughes’ *Playworker’s Taxonomy of Play Types* [22,23], and Rubin’s *Play Observation Scale* [24,25,26]. However, each of these typologies pose difficulties for the categorization of play types observed in outdoor settings.

In his 1992 book *Play and Playscapes* [21], Frost, a well-known play researcher, synthesizes play types proposed previously by psychologists Buhler (1937), Piaget (1962), and Smilansky (1968) to identify four types of “cognitive” play which were considered to emerge in parallel with a child’s intellectual development, outlining an evolutionary set of play types (See Table 1). Beginning with the earliest form of *functional play*, the simple, repetitive actions observed within the first two years of life, Frost’s categories reflect the types of play that emerge as a child develops cognitively. *Functional play* is then supplemented with the more goal-oriented activities termed *construction play*, followed by the pretend games and make-believe behavior of *symbolic play*, eventually culminating in *games with rules* which were meant to reflect the highest level of cognitive development, that is, play with other children that is bound and regulated by a set of agreed-upon rules [21]. Frost acknowledges that the earlier forms of play do not disappear from children’s play activities as they progress to more mature forms, but also reinforces the notion that these types of play emerge in a fairly fixed and linear sequence, and implies that symbolic and rule-bound play are more desirable for being more cognitively advanced forms of play. To aid in the categorization of play behaviors during observations of children, Frost developed the *Play Observation Form*, which allows observers to code behaviors as either a form of *cognitive play, other play,* or *non-play* (See Table 1). The form privileges the three types of cognitive play outlined by Piaget [27]—*functional*, *dramatic,* and *organized games*—which are then cross-referenced with three levels of social play based on the work of Parten [28]—*solitary*, *parallel,* and *group* play. While Frost proposes most behaviors will be captured within this cognitive-social play matrix, the form does allow the observer to acknowledge “other” forms of play as well—*exploratory*, *constructive*, *rough and tumble*, and *chase games*. Behaviors such as *unoccupied*, *onlooker*, *transition,* and *aggression* were identified as *non-play*. Frost’s *Play Observation Form* was largely intended to study the play behaviors of children under the age of 5 years in indoor settings, such as preschools or childcare facilities. While his proposed typology may still serve to help identify a child’s level of cognitive development through play, and can perhaps be utilized for categorizing outdoor play types at a very high level, with only three main categories of play and without clear integration of the “other” play categories it is not broad or defined enough to effectively capture the diversity of rich play activities children engage in outdoors.

Hughes’ taxonomy of play types was developed primarily as a guide for playworkers, “to enable those who worked with children to call similar playful routines by the same names” [29] (p. 97). Hughes identifies 16 different types of play (See Table 1), from *symbolic play* to *rough and tumble play*, *communication play*, *deep play,* and *recapitulative play* as a way of helping playworkers to identify and facilitate a broad range of play activities, and to help them understand what children gain from engaging in different types of play [22,23]. While the taxonomy has proved useful as a framework for playworkers and early-year educators, there are several factors which make it difficult to use as a framework for an observational tool. The largest hurdle is that the types of play outlined often overlap in their descriptions, or their distinctions are not necessarily observable. For example, Hughes outlines several play types which all describe activities where a child explores or takes on a pretend role or persona as part of an imaginary play scenario—*role play*, *socio-dramatic play*, *dramatic play*, and *fantasy play*. Each type of play allows the child to experiment with different ways of being, and to test their own capacity for taking on different roles or tasks, often utilizing elements in their environment as pretend props. However, these four play types are not necessarily mutually exclusive, and the distinctions between them are often not clear or robust enough to allow for consistent categorization by an observer. For example, a child observed pretending to be a bus driver “driving” a cardboard box “bus” could conceivably fall under more than one of these categories. In addition, some categories, while describing play experiences that can be extremely valuable for children, represent a set of behaviors or learnings that are not necessarily observable to another person. For example, *deep play*, which involves encounters that allow children to navigate risky conditions or conquer their fear of elements, such as heights or bugs, can refer to a largely internal process of “steeling oneself” to meet a challenge and may not manifest itself in any visible way, making it very difficult to observe in progress. To be fair, Hughes did not develop this taxonomy to serve as a measurement tool. The lack of any kind of hierarchical ranking of the play types is a valuable takeaway—that is, Hughes is not endorsing any one kind of play as more desirable than another, but a number of obstacles remain that make the taxonomy difficult to use in its current form as an observational framework for outdoor play.

Rubin’s *Play Observation Scale* [24,25,26] is likely the typology which has been used or adapted most often in recent studies to categorize types of observed play (see Table 1). Rubin builds on the four categories of cognitive play originally posed by Smilansky [30]—*functional play, constructive play, dramatic play*, and *games with rules*—but works to more explicitly nest these play categories within the social play hierarchy advanced by Parten [28] to describe the sequential evolution of social or peer participation among preschoolers [24]. Parten described how very young children moved from *solitary play* and *onlooker behavior* on to *parallel play* with peers around the age of 2 ½ to 3 ½ years, and eventually on to *associative* and *cooperative play* before the age of 5 years [28]. Rubin integrated these two common, long-standing play hierarchies into a single typology for describing children’s play [24], eventually expanding the play categories to include *exploratory play* and a generic “other” play type coded as *occupied* [25,26] and, similar to Frost, condensing the social play categories to *solitary, parallel,* and *group play*. Rubin’s *Play Observation Scale* (POS) was designed in part to be able to study children’s socio-emotional development, and to identify children who are socially withdrawn or aggressive and potentially at risk for future psychological difficulties [25]. While the POS is useful for studying children’s play behaviors, particularly indoors, like Frost’s observation protocol, it does not provide enough relevant categories, or distinction within categories, to capture the full range of common outdoor activities.

Another drawback of Rubin’s scale is the relegation of some commonly witnessed play behaviors to a “non-play” category, including *onlooking* or *exploratory behaviors, active conversation, transition,* and *rough and tumble play* (See Table 1). While Rubin eventually integrated *exploratory behaviors* as a legitimate play type in later versions, the other activities continue to be designated as *non-play*. An extensive review of the literature and our own field experience, however, supports the recognition of *onlooking behaviors, rough and tumble play,* and in some cases, *active conversation* as genuine play types or subtypes. Onlooking behavior, for example—where a child steps back from the main play event, occupying a safe but remote spot from which to watch others playing, is often an important bridge between episodes of involvement with others and should be acknowledged as part of the entire play cycle. *Rough and tumble play* is also considered to be non-play by Rubin, yet research has demonstrated that play fighting or wrestling, which takes place between smiling friends [2,31] not only helps to build children’s strength and dexterity, but is a valuable play activity for developing social awareness and competence, empathy, and a sense of cooperation, and may relate to pro-social behaviors in adulthood [1,2,32,33].

Rubin’s POS and other play typologies used in outdoor play research to date have largely been developed to study children’s behavior in controlled indoor environments. Parten, whose social play categories were adopted by Frost and Rubin, acknowledges that indoor environments were preferred for her research as “it was thought that elements might enter into outside play which did not exist in indoor play” [28] (p. 247) and these elements were likely to confound the research which contributed to her classification system. Rubin does not explicitly state that his scale was developed for observing indoor play, but many category descriptions describe what the play behavior “in the room” might look like or how to code behavior when a child “leaves the room” [24,25]. As a result, many outdoor play studies have found it necessary to adjust these typologies to account for the unique play activities afforded by outdoor environments. As suggested by Parten, this is in part because the indoor play-type categories were not well-suited to addressing play interactions that may only be seen in outdoor settings (e.g., interactions with wildlife, picking dandelions, watching clouds, climbing trees, or water play). The authors’ own attempts to fit observed outdoor play into the POS categories led to significant inconsistencies in field coding of behaviors across observers.

This difficulty in capturing the complex nature of outdoor play is reinforced by Rubin’s protocol which directs observers to only categorize the most dominant play behavior observed. When there is no one dominant form of play, the observer is instructed to “code up”, that is, select the category that represents the most mature social and/or cognitive category, according to a hierarchy set out by Rubin [26]. This hierarchical structure privileges one element of play over another, and so not only fails to holistically capture the outdoor play episode, but negates important elements of the child’s activity. This protocol may be suitable if the ultimate goal is to assess a child’s social or cognitive development (though there remains some debate over which play activities may be the most socially or cognitively mature [21]), but it is less useful for capturing the diverse and integrative nature of play afforded by an outdoor play environment.

The commonly used play typologies of Frost, Hughes, and Rubin each have drawbacks which diminish their utility both for categorizing outdoor play behaviors and as an observation tool framework for the less-structured nature of outdoor environments. Researchers interested in naturalistic observations of outdoor play therefore need a more relevant and nuanced typology and tool that can not only capture play behaviors, but also help assess whether an outdoor play space is providing environmental support for a diverse range of play activities and a diverse group of players. Another advantage of developing one standard set of outdoor play types is the opportunity to be able to consistently compare results across studies and sites, and to collectively build a larger evidence database around children’s outdoor play behaviors and environments.

The new *Tool for Observing Play Outdoors* (TOPO) presented here outlines not only a new typology of outdoor play, but a systematic field observational protocol which utilizes these play types to effectively categorize children’s activities in outdoor and naturalized play spaces. The TOPO can be used to evaluate both outdoor play behaviors, as well as the outdoor play environment itself. This observation tool can also be used to evaluate the play behaviors of children within any age group, and in both formal and informal outdoor play spaces.

## 4. Development of the Tool for Observing Play Outdoors (TOPO)

The development and refinement of the TOPO tool, and its embedded outdoor play typology, involved five distinct stages.

Nota bene: In outlining the TOPO framework, we use the term “play episode” to describe the play activity that is captured during a single designated observation period, and the phrase “play cycle” to refer to the entire period of play activity a child undertakes within a given play session, which can comprise multiple play episodes.

### 4.1. Phase 1

To develop a new typology specifically suited to observing outdoor play behaviors, we began with an extensive review of literature related to children’s play needs, the developmental role of children’s play, and the growing body of research related to outdoor play. We also performed a detailed search and evaluation of existing play typologies. This work informed a systematic re-examination of three different field data sets of play behaviors collected by the authors during previous behavior mapping studies of children’s outdoor play (using an adapted version of Rubin’s POS typology). We subsequently flagged instances where (1) the play type codes were not capturing the essence of the outdoor play activity being observed and (2) the play episode was not being coded consistently across observers. We then examined all flagged play events, looking for commonalities that might suggest new categories or subcategories; emerging categories were then tested through an iterative process of re-coding several behavior mapping datasets.

### 4.2. Phase 2

This phase focused on establishing the primary and sub-categories of play to be included in the TOPO. Building on insights from the literature review and the field data analysis, we also considered each existing play type from the Frost, Hughes, and Rubin typologies to consider whether it was relevant for describing activities that may be commonly observed during outdoor play, as well as whether the current description was capturing outdoor play in either too narrowly or too broadly a sense. For example, as noted previously with the Hughes typology, very similar types of play (such as role play and socio-dramatic play) were separated into distinct categories; while these types of play may have subtly different meaning for the child or their development, they would be very difficult to differentiate during observations. In these cases, these types were collapsed into a single category to describe the broader type of play. Our analyses of other typologies, paired with the literature review, indicated that several previously defined play types, at least in some form, remained relevant and valuable for categorizing outdoor behaviors, including *functional* or *locomotor play* (renamed as *physical play*), *dramatic* or *symbolic play* (renamed as *imaginative play*), and *games with rules* (renamed as *play with rules*); however, the parameters of each were re-evaluated and broadened or narrowed as necessary. For example, *constructive play* also remained relevant to outdoor activities, but was in fact too narrowly prescribed; exploratory activities which did not result in actual “construction” were being missed.

Four key breakthroughs in the development of the TOPO tool took place during this phase. First, it became clear that we could effectively tease out distinct subtypes of play within each broader play type that would not only allow for richer and more detailed analyses of play behaviors, but could also improve inter-rater consistency when coding. A second breakthrough was a reframing of both a broader and a more nuanced description of *exploratory play* and several distinct subtypes, which would include *constructive play* but not be limited to it. The exploratory interactions between children and their environments are significant developmental activities associated with key cognitive skills, including learning and information recall [9,34]. Rubin uses *exploratory play* to capture “focused examination of an object for the purpose of obtaining visual information about its specific physical properties” [25] (p. 4), while Hughes expands this slightly to include multi-sensory manipulation of an object in order to explore its properties and inherent play possibilities [23]. While wanting to incorporate both of these forms of interaction, we struggled with the overlap with definitions of *constructive play*. The *Exploratory Behavior Scale* developed by van Schijndel, Franse, and Raijmakers [35] proved to be a valuable preliminary framework for reconceptualizing this category and its subtypes. Van Schijndel and colleagues distinguished between *passive exploration*, where a child attends to or engages their environment but does not manipulate it, and *active manipulation*, where a child manipulates an object or the environment in an active and attentive manner. We incorporated these distinctions in exploratory play subtypes of *exploratory-passive* and *exploratory-active*. *Constructive play* was then reframed as an advanced subtype of *exploratory play*. *Constructive or construction play*, defined in turn by Rubin [24] and Frost [21] as the purposeful manipulation of objects in order to construct or create something, can therefore be conceptualized as taking active exploration one step further; the child explores how to manipulate objects in the environment toward some building purpose, such as constructing a fort, a water dam, or a stone tower.

The third major change was the removal of social play as a distinct play type or as an embedded feature of the TOPO tool. Social play often refers to the way a child is engaging with and communicating (or not) with peers during a play episode, such as whether a child is playing on their own or cooperatively with others [21,25,28]. Both Frost and Rubin’s observation tools have embedded categories of social play, where the social nature of each interaction is captured simultaneously with one of the cognitive play categories. However, in each tool, social play behaviors are not treated consistently; some social aspects of play are outlined as social play categories, while others are considered *other play* or *non-play,* such as *active conversation*, *onlooker behaviors*, and even *rough and tumble games* which have a significant social component [2]. Hughes treated social play as a distinct play type, covering play interactions where rules and criteria for social engagement are explored and negotiated [23]. While we believe the social nature of play is an essential characteristic of children’s play to capture and study, the TOPO acknowledges that all types of play can be characterized by the social interaction that takes place during, or even helps to define a play episode. As suggested by Burdette and Whitaker [8], all play with others involves some form of a social problem; children must constantly discuss and negotiate what and how to play. These social interactions help children to learn how to cooperate and compromise, build their capacity for flexibility and empathy, and give them practice regulating their emotions and desires [8]. Integral social exchanges are already embedded in several of the TOPO play types, including *play with rules* and some forms of *imaginative play*. We have therefore chosen not to treat the social context of play as its own play type, but rather as an essential element woven into every play episode which would be better and more consistently captured using a complementary social interaction scale. The choice of social or peer interaction tool should align with the goals of the study, as well as the age group of the players being observed. A full or modified version of Parten’s social play categories remains a helpful framework (See Table 1).

The final innovation during this development phase was the expansion of the observation protocol to allow for up to two play types (with subtypes as relevant or desired) to be recorded for each play episode. Conceptually, we felt this change would both more accurately capture the complex nature of children’s outdoor play behaviors and largely negate the need for a hierarchical structure in order to improve the reliability of field coding. This protocol change was subsequently tested in Phases 3 and 5.

At the completion of Phase 2, we had established an initial outdoor play typology (TOPO Version 1; See Table 2) comprised of eight primary play categories and 29 associated sub-types.

### 4.3. Phase 3

The third phase of development involved testing and refining of this initial proposed TOPO typology. To strengthen and refine the play types, we conducted three separate rounds of reliability analyses (two within Phase 3 and one within Phase 5). For each reliability round, the two authors used the TOPO to independently code an identical set of previously recorded outdoor play episodes from three different outdoor play sites. A minimum of 150 records of play episodes were selected at random from datasets of 800 or more episodes for each round. For each play episode, the researchers were given the option to designate up to two play types (Play Type 1 and Play Type 2), each with a primary play type and a subtype. For each reliability round, the codes from both researchers were compared and scored on the degree to which there was agreement. The researchers then reviewed all records where there were inconsistencies in coding; in some cases, this review led to the development of a new play type or subtype, whereas in other cases it led to the inclusion of greater detail within the play type descriptions for additional clarity.

The first reliability round assessed the play types in TOPO Version 1 (See Table 2) and yielded an overall agreement score of 0.746, revealing that the two researchers were coding the same two primary and subtypes approximately 75% of the time (See Table 3). However, while agreement for the primary and subtype categories for Play Type 1 were very high (94.1% and 84.3%, respectively; see Table 2), coding agreement for both primary and subtypes for Play Type 2 was fairly low (65.7% and 54.5%, respectively). [Note: there was no hierarchical distinction made between Play Types 1 and 2; the order in which the coders listed play types for an episode was not considered, only whether one or both of the play type codes from one rater matched one or both from the other].

Reflecting on the coding inconsistencies, the researchers made several key changes, the largest of which was a substantial revision of the *communication play* category (See Table 2). A review revealed that inter-rater reliability decreased significantly when observers captured some non-essential discussions happening among the players using the *communication play* category. For example, if two children were observed talking about the timing of their lunch break while collectively building a sandcastle, a coder might use the *communication play-social conversation* combination as one of the two designated play types. Similar to social play, many forms of observed communication were a valuable element of the play, but did not represent a distinct play type; the *communication play* code was also superseding other play types which better captured the essence of the play episode, as well as leading to significant coding inconsistencies. While we kept the *communication play* category within the typology (later expanded and renamed as *expressive play,* see Table 2), this play type was limited to episodes where the communication was a primary component of the play. To capture all other forms of communication taking place within or around an observed play episode, we developed a separate, complementary *Play Communication Typology* (PCT) which can be used in concert with the TOPO (see Appendix A for more details on the PCT).

Other changes included moving *transition* from a subtype of *physical play* (where it was initially placed because most transitions between play settings or in and out of the play space itself involved walking) to a subtype of *non-play,* which better reflected its character. The subtypes of *imaginative play* were also initially set out as *solo* and *group*; however, we realized that these subtypes reflected the social interaction of the play, which, as noted earlier, is better captured through a separate peer interaction measure. The subtypes would be more informative if they provided distinctions between common types of imaginative play. A review of the literature and other play typologies led to the development of three subtypes which represented fundamental, as well as observable differences in children’s pretend play activities—*symbolic, socio-dramatic,* and *fantasy play* (See Table 2).

A second round of reliability coding was performed by the two authors on Version 2 of the TOPO, yielding an overall agreement score of 78.0% (See Table 3). This overall score was an improvement from Round 1 and reflected greater agreement among Play Type 1 codes; however, agreement among the primary and subtype categories for Play Type 2 remained fairly low (68.1% and 56.0%). Removing play-related communication to a separate measure did significantly improve the level of agreement, particularly within the Play Type 2 codes, as raters were not having to use a second play type to capture communication taking place around the play event. When all play episodes were then coded by the two researchers separately using the new *Play Communication Typology*, inter-rater agreement was immediately very high, showing 86.4% agreement among coders. *Cohen’s kappa* calculated for play communication coding was 0.80, an extremely high level of agreement.

Refinements made after the second reliability round to address continuing coding discrepancies included more clearly defined subtypes; for example, more clearly distinguishing between behaviors that fall under *exploratory-active* versus *exploratory-constructive*. It was also decided that observers would try whenever possible to define a second primary play type (with associated subtype) for each observed play event, as most reliability errors were a result of one rater including a second play type when the other did not.

### 4.4. Phase 4

After these small revisions to Version 2, the TOPO typology was shared with an external panel of 10 researchers, play providers, and design practitioners who are intimately familiar with children’s outdoor play behaviors. Reviewers were asked to consider the typology in relation to their experience with outdoor play, and to answer a series of questions related to potential gaps or mischaracterizations within the play types, the value of a hierarchical framework, and the utility of the typology as an observation tool. These reviews yielded substantial insights which prompted changes and additions to the play types to provide additional clarity, to improve concept validity within subtypes, and to address gaps. These changes are outlined in TOPO Version 3 (See Table 2). One significant change was renaming and expanding *communication play* to *expressive play*, in order to better describe and include a broader range of expressive and artistic play behaviors common among children; *language play* and *artistic play* were added as additional distinct subtypes (and *social conversation* was simplified to *conversation*). A few other adjustments were made, including moving *nutrition* (eating and drinking activities) from a subtype of *restorative play* to a *non-play* subtype, and the addition of *onlooking* as a new subtype of *restorative play* (see below for more details). *Exploratory-passive* was renamed to *exploratory-sensory* to better reflect the intentional interaction with an object or environment using one or more senses (but without active manipulation). Distinct subtypes were also developed for *digital play* to reflect the different interfaces between digital elements and the physical environment that can occur in outdoor play spaces (see Table 2).

However, the largest insight arising from the expert review was the inability of the current typology to capture the playful interactions we often see outdoors between children and living things, such as plants, bugs, and birds. These interactions would be rightfully captured under *exploratory play*, but this play type was not sufficient on its own to reflect the essence or importance of these encounters with plants and wildlife that is often unique to outdoor environments. We subsequently expanded the original play type *stewardship*, which was meant to capture interactions that involved care of living things or the environment generally, to a new primary play type called *bio play*. More details on this new outdoor play type can be found below.

These changes were incorporated into Version 3, the final iteration of the TOPO tool (See Table 2).

### 4.5. Phase 5

For this last phase of development, a final reliability round was carried out by the two authors using TOPO Version 3 (See Table 2). Inter-rater agreement across all measures showed marked improvement (See Table 3). There was 100% agreement around the primary category for Play Type 1, and 99% agreement on its associated subtype. Agreement on Play Type 2 was improved from previous rounds, but still lower than that of Play Type 1. Coders agreed on the second primary play type 73.5% of the time, and 61.5% on the second subtype. Investigation of the coding discrepancies revealed that this lower agreement rate for Play Type 2 was primarily the result of two conditions: either one coder had decided not to use a second play type to categorize the play episode, or else the raters agreed on the second primary play type but selected different subtypes within the same category, such as one coding an episode as *exploratory-active,* while the other coded it as *exploratory-constructive*. Each minor disagreement was revisited to understand the reason for the inconsistency, and both the protocol and the subtype descriptions were further refined to minimize future discrepancies. As part of this work, the authors identified common intersections between play types that tend to be observed during outdoor play to be included as part of training materials for the tool; some of these intersections are outlined in Table 4. However, when we calculated the overall agreement for the primary play types, raters agreed 86.8% of the time, and overall agreement on all subtypes was 80.3%; to ensure that these generally high levels of agreement were not a result of chance, we calculated *Cohen’s kappa* coefficient for both primary and subtypes. The kappa value for agreement among primary play types was 0.80, demonstrating almost perfect agreement, and 0.75 for subtypes which reflects substantial agreement among raters [36]. The final version of the typology therefore produced sufficiently high levels of inter-rater agreement and the ease and reliability of coding was greatly improved from previous versions. Table 3 outlines the final iteration of the Tool for Observing Play Outdoors; detailed descriptions of each play type are outlined below.

### 4.6. Validation of the TOPO

While developing this typology, efforts were made to follow a course of action that would result in a valid and reliable tool that could be used consistently by multiple researchers and garner-consistent results across many types of studies. Cresswell and Poth outline nine strategies for effectively validating qualitative data and recommend that researchers utilize at least two of the recommended strategies in any given study [37]. Our development approach focused on five of those strategies: corroboration of evidence through triangulation, prolonged engagement, and persistent observation in the field, generating a rich, thick description, having a peer review or debriefing of the data and research process, and enabling external audits [37] (pp. 260–263). Below is a snapshot of how we aimed to fulfill each strategy:*Corroboration of evidence through triangulation*: The authors carried out an extensive review of literature related to play and children’s developmental needs, as well as a detailed examination of other play typologies. Results of these reviews were continually cross-referenced with previously collected field data of children’s play behaviors and feedback from experts; all types of data were examined for corroboration and refinement of emerging outdoor play types.*Prolonged engagement and persistent observation in the field:* Both authors had extensive experience collecting outdoor play type data from multiple outdoor sites both before and during the development of the outdoor play typology. Large data sets of outdoor play behaviors were also re-examined multiple times to ensure we had extensive knowledge of the full range of play behaviors observed in the field.*Generating a rich, thick description:* To provide transparency around its development, as well as to support transferability of the tool to multiple contexts, we produced detailed accounts of the process used to develop and test the tool, substantial descriptions of the play types, clear comparisons to previous play typologies, and outlined illustrative examples of each play type.*Having a peer review or debriefing of the data and research process:* The researchers jointly discussed the process and the findings on continual basis, and also met periodically with colleagues to discuss the development of the typology. These were informal sessions meant to confirm proposed development processes and discuss alternative strategies.*Enabling external audits:* Phase four consisted of an external audit where feedback on the typology (through a series of guided questions) was solicited from a panel of 10 professionals who had no connection to the study. This external review was extremely valuable for the validation process and led to several adjustments to the typology.

## 5. Outdoor Play Type Definitions

This section provides a fuller description of each of the nine primary play types in the TOPO (See Table 4). The first five play types largely correspond to categories of play also incorporated in some way in other play typologies—*physical play*, *exploratory play*, *imaginative play*, *play with rules,* and *expressive play*. The next three play types—*bio play*, *restorative play,* and *digital play*—are unique to the TOPO and represent newly derived categories which reflect either play behaviors that are particularly common during outdoor play, or else the changing nature of both play and play environments in the 21st century. The ninth and final category is a *non-play* category, which allows observers to code behaviors that are common to outdoor play environments, but which are generally not considered to be play activities. Details and examples of each primary play type and their designated subtypes are outlined in Table 4.

### 5.1. Physical Play

This play type captures outdoor play activity that is mostly physical in nature, where children’s use or testing of their bodies and their physical capabilities are integral to the play event. Sometimes categorized as *locomotor play, movement play,* or *functional play* in other typologies, this form of play allows children to explore their physical capabilities, make sense of their bodies, and develop increased strength, dexterity, agility, balance, and flexibility [2,3,9]. *Physical play* applies to behaviors such as running and climbing where children engage their large muscle systems, or play which involves finer motor movements and hand–eye coordination through the gripping, movement, and manipulation of small objects and tools [1,34]. This play type also captures vestibular play, such as balancing, spinning, or rocking; through such movements, children sense and explore their own position and movement in space, and develop a sense of equilibrium and postural control [38]; vestibular movements have also been linked to key developments in cognition and spatial behaviors [39]. Activities that involve playful physical contact, such as mock fighting or wrestling, are also captured as physical play. These activities, which take place “between friends who stay friends” [2] (p.89), are not actually acts of aggression as they can often be interpreted by adults, but rather an unappreciated type of play that helps children to develop social awareness and a sense of fairness and cooperation [2,21]. Such encounters can be so important to children’s social development that it has, at times, been categorized as a form of social play [32,40]. However, we capture these activities here as *physical play*, as they manifest primarily through physical contact.

Through physically embodied play, we use the movement of our bodies as a way of knowing, of understanding not only our own selves but our relationship to time, space, and other people [2]. While many of these *physical play* activities reveal themselves primarily through physical movement, we acknowledge that these actions often have associated cognitive, social, and emotional development components [9,34]. Jumping off of a high platform uses and builds physical capabilities, but to the child it can also be an embodied experience of gravity, and the result of conquering one’s nerves. Research on brain development confirms that “movement play lights up the brain and fosters learning, innovation, flexibility, adaptability, and resilience” [2] (p. 84).

*Physical play* captures a diverse range of physically involved activities, including for children of different ages and those with physical impairments or delays. *Physical play* activities can also be distilled into four distinct subtypes—*gross motor*, *fine motor*, *vestibular,* and *rough and tumble*. Detailed descriptions for and examples of these subtypes are outlined in Table 4.

### 5.2. Exploratory Play

This play type is used to describe playful interactions with the environment where the child is exploring or manipulating the properties of an object or the environment, either in a more sensory-based learning capacity or towards some child-established goal. Children are innately curious about their world and drawn to learning about it through interaction [2]. The degree of engagement and manipulation can vary from more passive, sensory-based interactions, such as rubbing a leaf or running their fingers through a pile of sand to experience their textures, to more active manipulation of the environment, like shoveling dirt into a pail, or to even more constructive activities where the child is actively building or forming something out of environmental elements, like a sand castle or a fort. It is through such exploratory engagements that children learn not only about the physical and spatial properties of their environment, but about their own capacity for manipulating objects or environments towards some intrinsic goal or problem [1,2,41]. The skills developed during exploratory play, such as sensory observation and an understanding of cause-and-effect relationships, are key to cognitive development; they can stimulate creative thinking, spatial reasoning, and problem solving, and are also considered central to math and science learning [2,8,35,42,43,44,45]. Several studies also suggest exploratory play with objects helps to lay the foundations for language development and literacy skills, particularly when combined with pretense [34]. The playful repetition of exploratory activities assists with the development of neural pathways in the brain which cement these movements and make them easier to recall in the future [1,2]. When it is valuable to understand the degree of engagement and level of environmental manipulation, *exploratory play* can be coded using the three subtypes—*sensory*, *active,* and *constructive* (See Table 4).

Exploratory play in the TOPO typology encompasses play activities that have been categorized elsewhere as sensory play, sensorimotor play, object play, mastery play, cause-and-effect play, construction or constructive play, and to some degree, creative play or deep play (See Table 4). It is important to note that this play type focuses primarily on children’s interactions with elements within their environment, not with other people; social interactions are captured either through other play types or separately with a social interaction measure.

### 5.3. Imaginative Play

This play type is for activities which involve any element of pretense, role play, or imagination. This includes pretending that a stick is a cell phone or a wizard’s wand, to taking on social roles such as a parent, chef, or bus driver, to more fantastical roles like commanding a spaceship or playing a super hero. Through imaginative or pretend play, children construct their own realities where they experiment with different roles, learn to solve problems, and in turn improve their mastery of their social and physical environments [21,46]. Children are inherently and universally drawn to creating and acting out pretend narratives, which has been shown to be “a key to emotional resilience and creativity” throughout the lifespan [2] (p. 87). *Imaginative play* is a way for children to safely experience and resolve social and moral situations, express their needs, develop empathy and trust, and act out potential solutions without negative consequences [1,2,21,46]. *Imaginative play* has also been significantly linked to the development of social, self-regulation, language and cognitive skills, as well as reading and writing capabilities [1,3,21,34,47,48]. Recent work is providing support for the notion that pretend play can increase children’s creativity levels and contribute to the development of emotion regulation [49,50]. Outdoor environments, which tend to exhibit more environmental variation and place fewer limits on children’s playful interactions, can be particularly conducive to cultivating children’s curiosity and sparking imaginative play [8]. Imaginative play is often heavily reliant on the availability of loose or manipulable materials in the environment that can serve as props to the pretense [21], which may be more plentiful or varied in outdoor environments.

*Imaginative play* can be further divided into several subtypes—*symbolic, sociodramatic*, and *fantasy* activities (See Table 4). *Imaginative-symbolic* describes play where a child takes some real object in the environment as a substitute for a pretend object or animates it in some way, such as imagining a block of wood to be a running car or bringing a toy dinosaur or doll to life. *Imaginative-sociodramatic play* is where a child imitates a social, domestic, or interpersonal role that they could experience as adults, such as playing parents or “house”, organizing or cooking a meal, or pretending to go to work. This form of play allows children to enter the world of adults, experiment with the social tools and roles of the culture, and safely try on different roles to understand their nature and appropriateness [1,9]. The “social scripts” which underlie many socio-dramatic scenarios promote not only social learning, but may indirectly promote reading comprehension and other forms of intellectual development [1] (p. 21). The third subtype, *imaginative-fantasy*, also describes role play but relates to scenarios that are not personal or domestic or likely to be experienced in real life; for example, when a child pretends to be a princess, wizard, a superhero, or becomes the dinosaur themselves. Role play where children integrate symbolic play with enacted narratives can be especially valuable during early childhood, supporting the development of emerging literacy and language skills [51,52]. Note that identifying the socio-dramatic and fantasy subtypes of *imaginative play* relies heavily on both contextual and verbal cues; while it may be obvious on the surface that the child is involved in some sort of pretend play, identifying the subtype often requires hearing the child’s narrative about their activity as a “parent” or “chef”, or observation of an associated environmental prop, such as a play oven or a superhero cape.

### 5.4. Play with Rules

This play type expands on previous play typologies’ use of *games with rules* to more broadly capture any play activity or game by two or more children where there is some agreed-upon framework of rules governing the activity. Children naturally seek to make sense of their physical and social worlds, and so are inherently drawn to rules and rule-based play from a very young age [34]. This kind of play may often take the form of conventional games, such as hide-and-seek, capture-the-flag, or baseball, where there is a pre-established set of rules that tend to be universally understood by players. However, it also includes more organically developed play games and scenarios where children invent, negotiate, and continually modify the rules of play; for example, a group of children may collectively decide to take on superhero or villain personas and establish an elaborate set of rules dictating how and where the superheroes can chase and lock up villains. While this play is also *imaginative*, the key element being captured by this play type is the process of negotiating and abiding by a set of play parameters.

Burdette and Whitaker remind us that rule-based play is inherently social, as it always requires “solving some form of a social problem, such as deciding what to play, who can play, when to start, when to stop and the rules of engagement” [8] (p. 48). In fact, children can spend most of their play time and energy developing, negotiating, and modifying play rules with their peers [34]. Rule-based play provides children the opportunity to test and learn about social and cultural practices [1], including the value of taking turns and listening to the perspectives of others [34]. In working through dilemmas and conflicts with others, which often includes controlling their own impulses and behaviors, children build emotional skills and capabilities that are critical to successful social relations during childhood and later in life, such as cooperation, flexibility, empathy, and self-regulation [3,8,21,34]. Engagements where children negotiate and collectively problem solve with peers has also been shown to support language development [3].

*Play with rules* can be divided into two key subtypes—*conventional* and *organic*. *Conventional* activities cover common, universally understood rule-based games, such as hide-and-seek or soccer, where players are agreeing to play by the set rules but little negotiation is needed to establish the initial rules of play. *Play with rules—organic* is intended to capture those play scenarios that have been developed together by children, often from scratch, and which continue to evolve as play progresses. *Conventional* rule-based play can actually morph into the *organic* form when players decide to significantly alter the universal rules during the course of play, such as adding a fourth base in a baseball game, or requiring that players circle the bases by running backwards. Table 4 provides a more detailed description and examples of these subtypes.

### 5.5. Expressive Play

This play type is used to characterize those play activities where some form of communication or expression is integral to the play activity. Children engage in many forms of *expressive play*, even from infancy. Very young children coo and babble, playing with sounds and vocalizations until, and even beyond the development of language abilities [1,9]. The interest in sounds, language play, and vocal performance evolves into singing songs, mimicry, telling rhymes, making up jokes, and reciting tongue twisters [1,21]. These expressive experiences have been linked to language development, and later, reading skills [1,9]. Children are also drawn to expression through other artistic and creative endeavors, such drawing, painting, sculpting, mark-making, as well as through musical, dance, and dramatic performances. Frost, Wortham, and Riefel suggest that school-age children particularly are “eager inventors and artists who demonstrate confidence and competence in their creative endeavors” [9] (p. 202). When they craft patterns in mud or paint pictures on paper, children are becoming aware that they can make something that is theirs alone and are developing an aesthetic appreciation for their environments [9]. Several alternative education programs, such as the Reggio Emilia school, believe that engaging in playful and artistic activities also helps a child to more deeply absorb, consolidate, and extend what they have learned from their observations and experiences [9,53].

This play category is therefore intended to capture a range of *expressive play* activities and can be further divided into four subtypes—*performance, artistic, language*, and *conversation. Expressive-performance* captures playful activities where a child is intentionally performing some way, such as singing, dancing, juggling, or playing music explicitly for the enjoyment of others. *Expressive-artistic* is used when a child is manipulating the environment specifically for an artistic, creative, or aesthetic outcome, such as arranging leaves in a pattern, drawing spirals or pictures in the dirt or sand, or using something in the environment to make playful noises. The *expressive-language* subtype describes activities where a child is playing with words and vocal sounds, including making up rhymes, jokes or chants, storytelling, or playful vocalizing. Finally, *expressive-conversation* is intended to categorize play where the primary engagement is talking to or verbally interacting with other children but which does not involve any pretend or role play; for other forms of verbal communication taking place during or alongside play, we recommend using the separate *Play Communication Typology* measure (see Appendix A). Many of these *expressive play* activities are often seen concurrently with *exploratory play* or *imaginative play* activities. More details and examples can be found in Table 4.

### 5.6. Bio Play

Children’s experiences in nature are important, as they can be linked to positive health and wellness benefits, cognitive development, and emotional connections to nature [54]. Theory and research suggest that children’s direct and unstructured encounters with wildlife are more developmentally significant than structured encounters mediated by adults [55]. Viewing animals in the wild, as compared to in captivity, can have longer lasting sensory impressions and contribute more to the development of an emotional affinity toward animals [56]. This strengthened impact might be explained by the knowledge that stronger emotions occurring at the time of an experience can lead to more vivid and long-lasting memories [57]. Natural interactions may be more exciting and thrilling for children than controlled exposures, and therefore, may denote a more significant opportunity to shape a child’s lasting memories. Childhood play in nature can also lead to lifelong preferences for natural experiences and pro-environmental behaviors [58,59].

In order to be able to capture these important interactions with nature, we have added a new category of outdoor play called *bio play* (drawing on the Greek root for “life”). This category can be used to record those significant moments when a child has focused their attention on a living plant or animal in the playscape. Although these experiences might also be recorded in another category, such as *exploratory play*, the significance of these natural experiences is profound enough to warrant capturing these interactions through a distinct play type.

Educators in the field of sustainable development believe that children can benefit from three aspects of environmental experience; the opportunity to be educated *about* the environment, to *experience* the natural environment, and to *act for* the environment [60]. Several subtypes allow for a further level of distinction among these environment interactions during outdoor play (see Table 4). The *bio-plants* combination is used to categorize playful interactions with living plants, from grass or flowers to shrubs and large trees. *Bio–wildlife* can be used to identify interactions with animals, from small bugs, butterflies and worms, to larger wildlife species like birds, fish, and mammals. A third subtype, *bio-care*, is designed to identify children’s play-based stewardship behaviors where the child acts with concern for the health of the environment, or demonstrates an appreciation or understanding of the value of nature and natural resources. This may include activities such as recycling waste, watering a plant, “planting” an acorn, or “rescuing” a caterpillar. Overall, the *bio play* category is useful for capturing the range of valuable interactions with living things which natural playgrounds can offer.

### 5.7. Restorative Play

The TOPO includes another new category of play referred to as *restorative play*. This category contains the subtypes *resting, retreat, reading,* and *onlooking.* Observations of play demonstrate that there are times when children will retreat from a play episode or hesitate to join other children engaged in play. There are many possible motives, including wanting to watch the other children play, avoiding conflict, or taking a break from a play cycle that required a sustained focus of attention. According to Kaplan, nature may contribute to attention restoration [61], which may play a significant role in the benefit of the outdoor play experiences for children. An outdoor play typology then, should include the ability to identify these and other restorative behaviors.

The restorative nature of resting and reading behaviors are fairly self-explanatory. The subtype *retreat,* however, requires more explanation; during a play cycle, children will often temporarily pull back from interactive play or seek out a small, safe place to take respite from more active or social play activities. Sobel and Hart have both written about children’s widespread use of “special places”—finding or building small-scale, somewhat enclosed spaces (e.g., dens, forts, treehouses, playhouses, or even cardboard boxes) where they often go to enjoy time alone or play with a friend [62,63]. Children have a tremendous need for both physical and psychological privacy with few opportunities to achieve it; in these “away” places, children explore a sense of separateness, as well as exercise a measure of control over their environment and interactions [63,64,65]. A recent study from Australia notes that over 85% of 8 to 9-year-old children expressed a high level of enjoyment from “hiding” outdoors, and 70% of these children enjoy using their recess time for resting and relaxing outdoors [66]. In another study, over a quarter of children sampled (aged 8–13 years) reported that they went to their favorite places after emotionally challenging events and for cognitive restoration and relaxation [67]. School children have been found to retreat from larger spaces in school yards, finding places on the edges which offered the freedom to play away from the crowds [68]. Children and youth often seek out and benefit from opportunities to retreat or rest from their daily experiences as part of their play cycle.

The *onlooking* subtype also describes a form of restorative play. Parten described a category of social play as *onlooker* [28], which has been used by researchers to describe children who may exhibit anxiety or reticent behavior which may prevent them from interacting with others [69], and which appears in the typologies of Frost and Rubin (see Table 1). It is common for children to demonstrate reticent behavior in which they hover on the edge, observing other players in a social circle; however, different forms of solitude may indicate different “meanings” for each child, depending on age, personality, and circumstances [69] (p. 135). Although this behavior is often thought to identify children who are shy and not as socially advanced, it is clear from the Coplan et al. study [69] that most children engage in some form of observer behavior during play group experiences. Largely a neglected element of play to date, it can be useful to understand how children use the environment for periods of restoration during a play cycle.

### 5.8. Digital Play

Any play typology developed in the 21st century must acknowledge the growing prevalence of play with and through digital devices, even in outdoor environments. Advancements in digital technology, particularly in the increased development and accessibility of portable digital devices for game play, such as smart phones, tablets, and game devices, has led digital play to comprise an increasingly large part of many children’s play activities [70,71]. While research related to digital play is still in its infancy, reviews of children’s play with computer or video games shows that there can be positive developmental outcomes, including knowledge acquisition, as well as cognitive and perceptual advances [34].

Not only is it becoming much more common to see children playing on some sort of digital device while in an outdoor play space, but digital technologies are also being embedded more and more in some form within outdoor play spaces themselves to provide novel ways to engage children in playful activities. For example, in the Darling Quarter Precinct in Sydney, Australia, an interactive digital art platform was installed in a large green space that allows visitors to use a touchscreen kiosk to digitally “paint” designs in light or else play over-scaled arcade games on the facades of the buildings surrounding the park. On a smaller scale, some outdoor play spaces are installing small digital sensors that trigger musical sounds when a visitor passes by, allowing players to compose their own “songs” by moving through the space. While still usually comprising just a small portion of observed outdoor play activities, we felt that a new outdoor play typology must include some ability to acknowledge and capture the growing popularity of digital play. Subtypes developed within this category help to distinguish whether the play is happening primarily on a device with no interaction with the physical environment (*digital-device*), or when a portable digital device is being used to create a digital-physical hybrid environment, such as through augmented reality games like *Pokémon Go* where the digital elements are overlaid on the physical environment (*digital-augmented*). A third subtype—*digital-embedded*—captures play that engages digital elements which have been integrated with the physical play environment itself through the use of sensors or interactive digital platforms. See Table 4 for more details and examples.

### 5.9. Non-Play

This final category is intended as a way to code behaviors which take place during an overall outdoor play cycle, but which are not considered to be “play”. Non-play activities include scenarios where a child is stopping to take care of themselves in some way, such as tying a shoelace, cleaning their hands, or taking a snack break. This category can also be used when the primary behavior observed is one of distress or aggression, where the child is disengaged entirely from play and either behaving in a distressed manner, such as crying or throwing a tantrum, or acting in an antagonistic way toward another child or adult, such as hitting, kicking, grabbing, or threatening them. *Non-play* can also be used to denote when a child is simply in the process of moving from one location to another, such as heading to the bathroom or moving to another play setting. This transition can sometimes be accompanied by some playful behavior, such as actively scanning the environment for birds or bugs as they move; in such a case, the behavior could also be categorized by another play type, such as *exploratory-sensory*. Finally, this category can be used to capture any other non-play activities observed, such as packing up belongings or some adult-directed tasks. These activities can be differentiated through the six subtypes—*self-care*, *nutrition*, *distress, aggression, transition*, and *other* (see Table 4).

## 6. Key Lessons and Innovations

The development and repeated testing of the TOPO yielded a few insights which are key to more effective use of the proposed outdoor play typology as an observation tool. The TOPO is intended to more effectively capture the diverse types of outdoor play that may be observed in a given setting and across a broad range of ages, and does not privilege one aspect of play as better or more developmentally advanced than another. To reflect this goal, we propose two particular innovations—eschewing any hierarchical framework of play types, and the use of up to two play types for each play observation to more fully and reliably characterize children’s outdoor play.

Many of the play typologies which have been used to study children’s play to date (e.g., Rubin’s *Play Observation Scale*, Smilansky’s cognitive play typology, or van Schnijdel et al.’s *Exploratory Behavior Scale*) were actually developed as scale tools which, by their nature, necessarily set up a hierarchy of play types, ranking play types relative to one another. Depending on the goal of the scale, such as capturing evidence of play that is considered to reflect a higher level of cognitive or social development, the developers privilege certain play types above others. Observers are directed to select the single play type which ranks higher on the scale. For example, if an observer witnessed a group of children playing an elaborate chasing game they collectively invented, during which they each also take on the role of wizards, by both Rubin’s POS and Smilansky’s play scales, the play episode could be categorized as both a *game with rules* and *dramatic play*. Since the observer can only choose one play type, they are directed to choose *games with rules,* as it is considered more cognitively and socially mature behavior by the developers. While such a protocol helps to improve reliability across different observers or when the play activity exhibits characteristics of more than one play type, these scale tools do not describe the full nature of the play activity.

Frost [21] reminds us that children’s play activities are often complex in nature, and it can be difficult to untangle an observed play episode and distill it into a single play type. In his own work, for example, Frost found that what he calls *functional* or *exercise play* was often intricately woven with *dramatic* or *constructive* activities [21]. Some studies have dealt with this issue by adding a *mixed play* category to be able to account for situations where the play reflects several of the other play types without one being dominant [19]. After years of field observations of play, the authors too continually struggled to accurately categorize play activities using a single play type, and in consequence, reported low levels of inter-rater reliability across multiple observers. When the goal is to capture the complex essence of children’s play activities or understand how well an environment supports a diverse range of play activities, establishing any kind of hierarchy within the play types is counterproductive. We also recognize that children do not necessarily develop physical, social, or cognitive skills along fixed, linear paths as once believed [9], and so a given play activity may not accurately reflect the developmental level or capacities of the child. Inclusive play advocates would also warn against developing a typology which values some play types as better or more advanced if they only reflect the developmental progression of fully able-bodied or neuro-typical children.

Through the development and testing of this new typology and tool, it became clear that unless the goal of a study is to record evidence of a specific type of play or interaction, to reliably and fully capture the nature of children’s outdoor play behaviors, the observation protocol must allow for a minimum of two different play types to be associated with a single play episode. The ability to record more than one play type largely negates the need for any hierarchy within the typology. It is also in the combination of the two play types that the more holistic spirit of the play is revealed. This protocol was tested in phases 3 and 5; when observers were encouraged to code up to two play types, inter-rater agreement increased substantially, and the combination of play types more faithfully reflected the play episode. The addition of a third play type did not significantly improve either the understanding of the play episode or the inter-rater reliability. Most third play types recorded related to the type of communication that was supplementing the play; when this verbal communication was removed to a separate measure using the *Play Communication Typology* (see Appendix A), the addition of a third play type was deemed largely unnecessary to the provision of a rich picture of the play episode.

## 7. Using the TOPO Tool

The TOPO tool can be effectively utilized for both person-based (i.e., extended observation over time of individual children) or place-based (i.e., systematic recording of the play behaviors of a range of children in a single setting) observations. When the goal is to assess the quality of the environment for supporting a diverse range of play types, a place-based protocol is more effective. For these studies we recommend a systematic scanning protocol where an individual child is observed for approximately 15 seconds, and then up to two play types (and subtypes if possible) which best capture the essence of the play episode are selected. When a play activity could be categorized under more than two play categories, observers are encouraged to choose the primary play types which best reflect the spirit of the play behavior, and to privilege all play categories over non-play types. See Appendix B for a sample protocol and template for using the TOPO for field observations.

Analyses of both outdoor play behaviors and environments can also be expanded by using the TOPO in combination with other measures designed to capture elements of play activities, depending on the goals of the study, such as: peer interaction, play communication, activity intensity, level of risk, use of loose parts, and wildlife interaction. Ideally, outdoor play activities are child-initiated and led, but some forms of play or some types of play environments may involve greater levels of adult direction or instruction; it can be helpful to use a supplemental measure to capture the level and form of adult interaction related to play activities. For a detailed field observation and behavior mapping protocol which combines outdoor play types with additional select measures to develop a comprehensive picture of children’s outdoor play activities, see this article by Cox, Loebach, and Little [72].

While the TOPO is primarily designed as a field observation tool, the embedded typology could also be used as a valuable framework for diverse analyses of outdoor play. The typology is also still highly suitable for observing children’s play in indoor settings, and in fact may add additional value due to the inclusion of newly defined play types. Unlike earlier indoor play type scales, the emerging categories of *bio play* and *digital play* in the TOPO can now acknowledge these increasingly common interactions indoors, such as school children taking care of classroom pets or plants, or children playing games on phones or tablets. While the proportion or intensity of some play types, such as physical play, may differ from outdoor settings, the TOPO can still effectively capture the range of play taking place (or not) in the indoor play spaces of schools, child care centers, community recreation facilities, and other indoor environments. Unlike previous play type tools, the ability to utilize the TOPO for both indoor and outdoor play also now allows for direct comparisons of play behaviors taking place in each type of setting, highlighting the differences in the support for play provided by both types of environments. The inability to use a single unmodified tool to effectively assess and compare both indoor and outdoor play behavior has been a significant gap in the field to date.

### Collapsed and Expanded Versions

Based on our own field experience and feedback from expert reviewers, we know that it can sometimes be difficult to utilize a large or complex observation tool under field conditions. Time is usually of the essence, as researchers aim to capture as many real-time play episodes as possible. A more complex tool can also require additional training time to minimize inconsistency between observers. However, some research studies or environmental evaluations require or desire a finer level of distinction between play behaviors. The TOPO tool has therefore been developed so that it can be utilized in either a collapsed or expanded version; researchers can choose which version best supports their research or evaluation goals, or which aligns with the time and resources available for field observations.

The collapsed version, TOPO-9, only utilizes the nine primary play types to categorize play activities. This version may be better suited to studies attempting to collect substantial data in the field or which have a limited amount of time for observations or training. The collapsed version might also better serve observations on larger or more complex study sites, when the play settings are large, in busy areas, or where it is difficult for an observer to get close enough to hear and see finer details of the play episode; the ability to make distinctions between subtypes often requires additional contextual clues that may only be apparent when the child can be heard or viewed up close. For example, a child pushing a large block through a pile of dirt can be coded from afar, as *exploratory-active* and *physical-fine*; the observer must be closer to hear them making car engine noises to know that they are imagining the block to be a car, now allowing the play episode to also be coded as *imaginative-symbolic*. Observers may in fact need to use the collapsed version when the study site does not allow for close, unobtrusive observation. For cases when only a high-level understanding of the diversity of play activities taking place in an environment is also needed, the TOPO-9 version would be sufficient.

The expanded version of the tool, TOPO-32, draws on the 32 associated subtypes to designate primary-subtype combinations for each observed play episode. This expanded tool allows observers to capture a much more detailed and nuanced picture of play behaviors taking place or being supported in a given environment. While requiring a bit more time for training and field collection, this version is well-suited to play environments that have several smaller-scale play settings or which allow for close but unobtrusive observation. The TOPO-32 provides a richer dataset which can be used to more thoroughly explore differences in play behavior by age, group size, gender, or other child characteristics, as well as reveal more details of how environmental features and conditions are supporting or limiting play.

While observation should be as unobtrusive as possible when using either version of the tool, the position of the observer should be close enough to see and hear details of the child’s play, especially when using the expanded TOPO-32. As Pellegrini suggests, to more fully understand children’s activities “observers might attend more closely to what children actually say and do during… play” [73] (pp. 571–572). The details which emerge through close observation are key to accurate and effective behavior coding.

## 8. Conclusions

This paper outlined the development and use of a new observational framework for studying child-initiated outdoor play, as well as the environments which support diverse play opportunities for children. Testing of the *Tool for Observing Play Outdoors* demonstrated qualitative validity, as well as a high level of reliability across trained observers. While our own primary interest was in preparing a tool which could effectively evaluate the play environment itself, the TOPO can easily be used for a range of studies of children’s outdoor and indoor play behaviors, as well as child health and development, particularly when paired with other complementary measures of play and interaction. This new tool represents a significant advance in the ability to fully and effectively capture children’s unique outdoor play behaviors, and to evaluate the quality of the environments which afford them. Use of this tool to study and plan outdoor play environments can help us provide more diverse, high-quality play settings that will support the healthy development of children across the spectrum.

Next steps in the development of the tool will include widespread testing of the TOPO in a diverse range of outdoor play spaces, from traditional outdoor playgrounds to children’s gardens, play trails, and nature play spaces. Observations using the TOPO will be triangulated with data from surveys and/or interviews with children, parents, and staff members (as applicable) to confirm that the tool is effectively and accurately capturing children’s outdoor play activities. This testing will further strengthen the validity and reliability of the tool, and confirm its utility for observations in a wide range of outdoor play environments.

The authors are also developing additional materials and resources to support TOPO training for observers, as well as its use in diverse environmental settings. We encourage play researchers and practitioners to contact us for more information related to training in or use of the TOPO.

## Figures and Tables

**Table 1 ijerph-17-05611-t001:** Play typology comparison.

Parten (1932)	K. Buhler (1937)	Piaget (1962)	Smilansky (1968)	Frost (1992)	Hughes (1996; 2002)	Rubin (2001; 2008)
*Social play*	*Cognitive play*	*Cognitive play*	*Cognitive play*	*Cognitive play*	*Play types*	*Cognitive*
Unoccupied behavior	Functional games	Practice games	Functional games	Functional play	Symbolic play	Functional play
Onlooker behavior	Construction games	Symbolic games	Construction play	Dramatic play	Rough and tumble play	Constructive play
Solitary play	Make-believe games	Games with rules	Dramatic play	Organized games	Socio-dramatic play	Exploration *
Parallel play	Collective games		Games with rules	*Social play*	Social play	Dramatic play
Associative play				Solitary	Creative play	Games with rules
Cooperative play				Parallel	Communication play	Occupied
				Group	Dramatic play	*Social play*
				*Other*	Locomotor play	Solitary
				Exploratory	Deep play	Parallel
				Constructive	Exploratory play	Group
				Rough and tumble	Fantasy play	*Non-play*
				Chase games	Imaginative play	Unoccupied behavior
				*Non-Play*	Mastery play	Onlooker behavior
				Unoccupied	Object play	Transition
				Onlooker	Role play	Active conversation
				Transition	Recapitulative play	Uncodable behavior
				Aggression		Out of room
						Adult interaction/conversation
						*Double coded behaviors*
						Aggression
						Rough and Tumble
						Hovering
						Anxious behaviors

* Exploratory behaviors were considered “non-play” in 1989 version.

**Table 2 ijerph-17-05611-t002:** Iterations of the Tool for Observing Play Outdoors (TOPO).

TOPO Version 1		TOPO Version 2		TOPO Version 3	
**Locomotor play**	*Gross motor*	**Locomotor play**	*Gross motor*	**Physical play**	*Gross motor*
	*Fine motor*		*Fine motor*		*Fine motor*
	*Vestibular*		*Vestibular*		*Vestibular*
	*Rough & Tumble*		*Rough & Tumble*		*Rough & Tumble*
	*Transition*				
**Exploratory play**	*Passive*	**Exploratory play**	*Passive*	**Exploratory play**	*Passive*
	*Active*		*Active*		*Active*
	*Construction*		*Construction*		*Construction*
			*Artistic*		
**Imaginative play**	*Solo*	**Imaginative play**	*Symbolic*	**Imaginative play**	*Symbolic*
	*Group*		*Socio-dramatic*		*Socio-dramatic*
			*Fantasy*		*Fantasy*
**Play with Rules**	*Formal*	**Play with Rules**	*Formal*	**Play with Rules**	*Conventional*
	*Informal*		*Informal*		*Organic*
**Communication play**	*Peer-Social*	**Communication play ***	*Performance*	**Expressive play**	*Performance*
	*Adult-Social*		*Social conversation*		*Artistic*
	*Play*				*Language*
	*Environment*				*Conversation*
	*Cowabunga*				
	*Instructive/Lesson*				
	*Self-talk*				
	*Care*				
**Performance play**	*[no defined subtypes]*	**Digital play**	*[no defined subtypes]*	**Digital play**	*Device*
					*Augmented*
					*Embedded*
		**Stewardship**	*[no defined subtypes]*	**Bio play**	*Plants*
					*Wildlife*
					*Care*
**Restorative play**	*Resting/Sitting*	**Restorative play**	*Resting*	**Restorative play**	*Resting*
	*Retreat*		*Retreat*		*Retreat*
	*Reading/writing*		*Reading*		*Reading*
	*Eating/Drinking*		*Eating*		*Onlooking*
**Non-play**	*Self-care*	**Non-play**	*Self-care*	**Non-play**	*Self-care*
	*Distress*		*Distress*		*Nutrition*
	*Transition*		*Transition*		*Distress*
	*Other*		*Other*		*Aggression*
					*Transition*
					*Other*

* Most Communications subtypes were moved to the separate *Play Communication Typology* (See Appendix A).

**Table 3 ijerph-17-05611-t003:** Reliability rounds summary.

	Inter-Rater Agreement						
Reliability Round	Play Type 1 Primary	Play Type 1 Subtype	Play Type 2 Primary	Play Type 2 Subtype	Overall IRR	Overall: Primary Play types	Overall: Play Subtypes
Round 1 (Phase 3)	0.941	0.843	0.657	0.545	0.746	0.799	0.694
Round 2 (Phase 3)	0.922	0.901	0.681	0.56	0.78	0.819	0.74
Round 3 (Phase 5)	1	0.991	0.735	0.615	0.835	0.868	0.803
						*kappa: 0.80*	*kappa: 0.75*

**Table 4 ijerph-17-05611-t004:** TOPO outdoor play types summary.

Primary Play Type	Play Subtype	Description	Examples	Common Intersections with other Play Types	Comparable Categories from Other Typologies
**Physical Play**	*Gross motor*	activities that utilize large muscles and/or require whole body movement; not just casual movement of an object, but activities which might tax muscles or help to improve gross motor skills; can also include large muscle activities that require hand/eye coordination	climbing, running, throwing/catching, lifting, carrying heavy loads, crawling, swinging an item such as a bat or branch, jumping, kicking, or riding a bicycle	*play with rules:* as in chasing games	functional play; locomotor play; exercise play; movement play; deep play
	*Fine motor*	activities that involve the use of smaller muscle movements and hand/eye coordination, or which help develop finer motor skills; can include picking up or manipulating small objects in the environment	using a spoon or stick to stir water or mud, using a small shovel to scoop sand, picking up leaves, balls or other small loose parts	*exploratory-active*: as in stirring a bowl of dirt and water to make mud	functional play; object play
	*Vestibular*	activities which test or improve a child’s sense of balance and/or reinforce their relationship to the earth; usually involve movement of the head or quick movements in multiple directions	balancing, spinning, twirling, sliding, rolling, rocking, or hanging upside down; play activities could include balancing on a log or bridge, going back and forth on a glider or see saw, rocking in a chair or hammock, riding a swing, doing somersaults or cartwheels, walking on their hands, skipping, using monkey bars or hanging from a tree branch		functional play; locomotor play; exercise play; movement play; deep play
	*Rough & Tumble*	engagement in playful or mock fighting or wrestling “between friends who stay friends” [2] (p. 89) or more broadly playful physical contact such as tickling. Note: needs to be distinguished from actual aggression, which is not meant as a playful exchange between friends and is categorized within OPOS as *non-play*	play fighting, wrestling, tussling, tumbling, tickling, “sword” fighting	*imaginative*: as in play fighting within a superhero scenario	social play; play fighting
**Exploratory Play**	*Sensory*	playful but primarily passive (i.e., non-manipulative) exploration of an object or environment, often through one or more senses; includes interactions where the child appears to be receiving sensory information about the object or environment where their attention is focused, such as rubbing a plant part to feel its fuzziness or roughness; also includes when a child is walking through an environment but clearly taking in or exploring the setting	rubbing a leaf blade, splashing feet through flowing water, searching the ground for bugs or flowers, running mud through their fingers, petting an animal	*bio play-plants* or *bio play-wildlife*: as in playing with plant or animal life	object play; exploration
	*Active*	playful activities that involve active manipulation of an object or the environment where the child is paying attention to the outcome of the action, movement, or interaction These activities may have some goal, such as filling a pail with shovelfuls of sand, but where the child is not necessarily building or constructing something.	shoveling sand into a pail, digging a hole in a pile of gravel, floating a boat down a stream, collecting rocks and leaves, using a tree branch as broom to sweep dirt, driving a toy truck through mud, pulling the petals off a flower, stirring water and dirt together to make mud	*imaginative*: as in a “parent” baking a mud “pie” to serve to family for dinner	object play; sensory motor; cause and effect play; construction play; mastery play
	*Constructive*	activities where the child is manipulating objects in or the environment itself for the purpose of physically building or construct something, or else the thoughtful destruction or taking apart of something; includes when a child is playing with or putting together any kind of puzzle.	using rocks to build a dam in a stream, using loose parts to build a fort, piling rocks up to build a pyramid, piling up sticks to build a “fire”, arranging crates to define a “house”, moving pylons to create a racetrack	*physical play* types: as in picking up or manipulating small or large objects as part of (de) construction activities *imaginative-play* types: as in constructing a tower to use in their role as prince or princess	construction play; constructive play; mastery play; cause and effect play; creative play; deep play
**Imaginative Play**	*Symbolic*	where the play involves using an object, action or idea in the environment as a symbol for something else. This pretense is essentially embedded in the other two imaginative subtypes, but this is to be used when there is no observable evidence of *socio-dramatic* or *fantasy play* elements	using a piece of wood to symbolize a person, or a piece of string to symbolize a wedding ring, or a banana to play the role of a phone	*exploratory-active* or *exploratory-constructive*: as in turning mud into “cake batter” or constructing a “house” out of cardboard boxes	creative play; dramatic play; symbolic play
	*Socio-dramatic*	where the imaginative play involves playing or trying out typical social, domestic or interpersonal experiences or roles they may experience as adults	playing “house”, going shopping, pretending to be parents, organizing or cooking a meal, pretending to have a family fight	*exploratory-active* or *exploratory-constructive*: as in pretending a plate of stones is “supper” for their children or arranging logs to form the outline of their “house”	symbolic play; dramatic play; role play; creative play; fantasy play; role play
	*Fantasy*	where the imaginative play involves performing or playing with situations that are not personal or domestic, or enacting something that is unlikely to occur in real life.	playing characters from Harry Potter, pretending to be a princess, a wizard, an animal, or a space pilot	*exploratory-active* or *exploratory-constructive*: as in pretending sticks are wizards’ wands, or a line of wood stumps as their castle moat	symbolic play; dramatic play; creative play; fantasy play; role play
**Play with Rules**	*Organic*	two or more kids are agreeing to play or challenge each other in a certain way, where they develop, negotiate and even change the rules as they go	developing a game where superheroes chase villains, or racing toy cars down a plank, or seeing who can climb highest in a tree	*physical play types*: as in running, wresting, or spinning as part of a made up game or challenge	games with rules; social play; locomotor play
	*Conventional*	two or more kids playing games that have common, universal or well-known rules that the players understand before commencing. Note: if the play morphs into an activity where the rules are changed or renegotiated so they no longer follow the traditional rules of play then in become *play with rules—informal*	soccer, baseball, tag, capture the flag, hide and seek	*physical-gross*: as in running to play soccer or tag	games with rules; social play; locomotor play
**Bio Play**	*Plants*	where a child observes, discusses, or interacts with a living plant	picking a fruit, closely examining a leaf or flower, or exploring or commenting on some characteristic of the vegetation	*exploratory-active:* when they manipulate portions of the plant, or *exploratory sensory:* when they comment that a leaf “feels fuzzy”	
	*Wildlife*	when a child is keenly observing or interacting with wildlife in the same environment including animals, birds and bugs (that are not a domestic pets)	catching small animals or bugs such as fireflies, moths, frogs, or crawdads; closely observing a bird or a turtle; poking a stick in the water to watch frog eggs wobble; looking under a log to see bugs	*exploratory-active:* when they pick up a bug, or manipulate the environment to better see an insect; *exploratory sensory*: as in when they are actively watching a bird, bug or animal	
	*Care*	a child acts in a way that demonstrates care or stewardship of the environment, or an appreciation of nature	watering a plant or planting an acorn, building a home for a turtle, rescuing a caterpillar that is crawling along a pathway, picking up a piece of litter and placing it in the recycling bin	*exploratory-active:* when they are filling a pail in order to water a plant, or gathering grass to feed a caterpillar	
**Expressive Play**	*Performance*	intentionally performing for others in some way	includes singing, drama/acting, dancing, playing music, juggling, or even hamming it up for the entertainment of others		musical play; creative play; symbolic play; semiotic play
	*Artistic*	manipulating the environment specifically for an artistic, creative or aesthetic outcome; includes mark-making and drawing	arranging leaves in a pattern, drawing spirals or pictures in the dirt or sand, painting or drawing pictures, making a pattern in the mud with footprints, or a sculpture out of sand	*exploratory-active* or *exploratory-constructive*: as in building of a patterned pyramid out of colored blocks, or designing a castle complex out of sand	creative play; symbolic play; semiotic play
	*Language*	activities involving the playful use or testing of sound, words and/or language	making up rhymes or poems, singing to themselves or with others, chanting, making up/telling jokes or stories,	*exploratory-active* or *exploratory-constructive*: as in using a blade of grass or branch to make a whistling sound, drumming on a pot with a spoon	storytelling; narrative play; communication play; musical play; recapitulative play; semiotic play
	*Conversation*	activities where the primary playful interaction is social conversation with other children or adults but does not involve any role play, is not supplemental to the play, or fall under other *expressive play* subtypes	includes two or more children sitting around a stump circle talking about a mutually attended event, or discussing their day with a parent	*restorative-resting* and/or *non-play nutrition*: as is a small or large group talking together while sitting in the shade or eating a snack	communication play; active conversation
**Restorative Play**	*Resting*	includes activities where a child is clearly taking a mental and/or break or rest	includes sitting, laying down, daydreaming, talking quietly to themselves (not paired with another play activity), or even quietly staring into space	*non-play nutrition*, *exploratory-sensory* or *restorative-onlooking*: as in sitting in the shade while taking a water break, or visually exploring the environment or other children while resting	
	*Retreat*	where a child has removed themselves to a small, controlled space; may include the ability to look out and watch others	includes crawling into or watching out from a fully or semi-private fort, den or other enclosed space	*exploratory-sensory* or *restorative-onlooking*: as in sitting in a small fort peering out at the environment or other children nearby	
	*Reading*	when a child is reading or writing for pleasure, or listening to others or music	includes reading a book, listening to another person telling a story or reading to them, or listening to music	*exploratory-sensory*: as in listening to a naturalist while they exhibit a live animal	exploration
	*Onlooking*	where a child deliberately steps back from nearby play for a period of observation rather than interaction; may just precede or follow play with others	when a child is sitting or standing apart while clearly watching others play or interact nearby	*restorative-resting* or *exploratory-sensory*: as in sitting on a log bench watching and listening to other children playing in a nearby setting	onlooker behavior; hovering
**Digital Play**	*Device*	where child is playing with or on a digital device with no interaction with the real world/physical environment	includes play games on a phone, tablet or portable game device, or listening to music through a device	*restorative-resting* or *restorative-reading*: as in sitting on a park bench playing a game on a phone, or laying on a blanket listening to music	communication play
	*Augmented*	when a child is using a digital device to mediate or augment their interaction with the physical world	includes playing Pokémon Go or other augmented reality game, or using their phone to read information transmitted through QR codes in the environment	*play with rules:* as in playing an augmented reality game with peers in the space	
	*Embedded*	when the child is interacting with digital prompts or devices embedded in the real world/physical environment without a personal digital device	includes activating digital sensors in the environment to hear sounds or see light displays, playing a digital instrument embedded in the environment, playing with an interactive digital screen in the environment	*expressive-performance or play with rules:* as in playing musical sounds via a digital instrumental device embedded in the space or playing a digitally-embedded game with friends	
**Non-Play**	*Self-care*	when a child is engaged in an activity meant to take care of themselves or their appearance; can include helping a friend or sibling to do these activities	includes taking off socks and shoes, tying shoelaces, tucking in shirt, or cleaning hands		transition
	*Nutrition*	when a child is taking a break to eat or drink	eating lunch or a snack, taking a drink of water		transition
	*Distress*	when a child is disengaged from play, and exhibiting signs of distress	crying, throwing a tantrum, throwing objects in frustration		anxious behaviors
	*Aggression*	refers to non-playful, agonistic interactions with another child or adult	includes hitting, kicking, grabbing, pinching, scratching, threatening		
	*Transition*	where the primary activity is non-playful movement from one space or point to another, and there is little to no active engagement or exploration of the environment	includes walking or running in or out of the play space, or from one play setting to another	*exploratory-sensory*: as in walking to the bathroom, but also actively visually exploring the environment as they go	transition
	*Other*	other types of observed “non-play” activities; can include “chores” or clean up work, especially if directed by an adult and not initiated by the child	includes picking up litter and placing in garbage bin, putting away play materials or gathering belongings when getting ready to leave the play space		transition

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
