# Peer review of "Tool for Observing Play Outdoors (TOPO): A New Typology for Capturing Children’s Play Behaviors in Outdoor Environments"

_ijerph, 2020, doi:10.3390/ijerph17155611_

Round 1

Reviewer 1 Report

This is an excellent paper: describes a critically important topic and is well written. Thank you for the chance to review. I fully support its publication. I noted a few typographical errors that a copy editor will detect readily. The only thing I wished had been included was a plan for further study that includes validity testing.

Author Response

Thank you so much for your review and comments.  Please see attached file for our detailed response.

Reviewer 2 Report

Very innovative and sound research article on the process of developing a tool to observe children's outdoor play.  There is good evidence in the review of the literature to support the development of the TOPO.  The developmental phases were well documented with adequate information about the process and how reliability was achieved. The 10 outside reviewers increased credibility and added important aspects to play typologies to make a more robust tool, however, details about the coders or data collectors is missing. While there is a section dedicated to using the TOPO tool, it was general and not specific enough which makes the tool difficult to utilized in future studies. Additional information about how this tool could be used in interior environments would be valuable information (although it is understood that your goal was to create a tool for outdoor environments).  

Author Response

Thank you very much for your review and comments.  Please see the attached file for our detailed response.

Thank you.
